# Effect of COVID-19 Vaccination on the In-Hospital Prognosis of Patients Admitted during Delta and Omicron Waves in Italy

**DOI:** 10.3390/vaccines11020373

**Published:** 2023-02-06

**Authors:** Rossella Cianci, Laura Franza, Giulia Pignataro, Maria Grazia Massaro, Pierluigi Rio, Antonio Tota, Francesca Ocarino, Marta Sacco Fernandez, Francesco Franceschi, Antonio Gasbarrini, Giovanni Gambassi, Marcello Candelli

**Affiliations:** 1Department of Translational Medicine and Surgery, Fondazione Policlinico Universitario A. Gemelli, IRCCS, 00168 Rome, Italy; 2Emergency Medicine Unit, Università Cattolica del Sacro Cuore, Fondazione Policlinico Universitario A. Gemelli, IRCCS, 00168 Rome, Italy

**Keywords:** COVID-19, vaccination, emergency medicine

## Abstract

All-cause mortality related to the SARS-CoV-2 infection has declined from the first wave to subsequent waves, probably through vaccination programs and the availability of effective antiviral therapies. Our study aimed to evaluate the impact of the SARS-CoV-2 vaccination on the prognosis of infected patients. Overall, we enrolled 545 subjects during the Delta variant wave and 276 ones during the Omicron variant wave. Data were collected concerning vaccination status, clinical parameters, comorbidities, lung involvement, laboratory parameters, and pharmacological treatment. Outcomes were admission to the intensive care unit (ICU) and 30-day all-cause mortality. Overall, the final sample included 821 patients with a mean age of 62 ± 18 years [range 18–100], and 59% were men. Vaccinated patients during the Delta wave were 37% (over ¾ with two doses), while during the Omicron wave they were 57%. Vaccinated patients were older (68 vs. 57 years), and 62% had at least one comorbidity Admission to the ICU was 20%, and the mortality rate at 30 days was 14%. ICU admissions were significantly higher during the Delta wave than during Omicron (OR 1.9, 95% CI 1.2–3.1), while all-cause mortality did not differ. Unvaccinated patients had a higher risk of ICU admission (OR 2.0, 95% CI 1.3–3.1) and 30-day all-cause mortality (OR 1.7, 95% CI 1.3–2.7). Results were consistent for both Delta and Omicron variants. Overall, vaccination with at least two doses was associated with a reduced need for ICU admission. Even one shot of the vaccine was associated with a significantly reduced 30-day mortality.

## 1. Introduction

The SARS-CoV-2 pandemic continues to have a high impact on hospitals worldwide. Constant mutations of the spike protein of the virus have generated variants to elude the protection offered by vaccines. At some point during the pandemic, the Omicron variant has overtaken the Delta variant due to its higher infectiousness [1]. 

The effectiveness of vaccination in preventing the spread of the virus and limiting the severity of the disease has consistently been documented. However, such protection tends to fade away with time from vaccination, and it is also dependent on patient characteristics, vaccine types, and immunization regimens [2]. These factors have created concerns in the general public, who have started questioning whether the first vaccines were still useful against these new variants [3].

Overall, vaccines seem to afford weaker protection against infection by the Omicron variant compared to the Delta variant. However, a triple dose of mRNA vaccine or a natural SARS-CoV-2 infection in patients who had already received two doses of vaccine seems able to protect also against Omicron variant BA.1 [4]. Indeed, in people older than 60 years, a booster dose has been shown to confer greater protection against both infection and hospitalization compared to just two doses [5]. A booster dose of mRNA vaccines increases the levels of antibodies and protection against COVID-19 disease [6]. Yet, a matched, case-control study indicated that a booster dose is more protective against Delta variant than it is against Omicron [7].

Thus, our study aimed to evaluate the impact of mRNA vaccination on the in-hospital prognosis of patients with the SARS-CoV-2 infection who were admitted to an academic medical center during the Delta and Omicron waves.

## 2. Materials 

This retrospective monocentric observational study was conducted in an Italian third-level academic medical center (Fondazione Policlinico Agostino Gemelli—IRCCS in Rome). The current analysis includes patients who presented to the Emergency Room between 1 July and 30 September 2021 during the SARS-CoV-2 Delta variant wave, and in January 2022, during the SARS-CoV-2 Omicron variant wave. For each patient, trained medical staff collected data on vaccination status (both SARS-CoV-2 and influenza virus), clinical characteristics, comorbid conditions, severity of lung involvement (presence of pneumonia on plain chest radiograph, pulmonary embolism, need for oxygen supplementation), laboratory parameters, and pharmacological treatment. Only patients older than 18 with no known history of COVID-19 positivity were included in the study. Data were extracted from electronic health records, and the study was approved by a local ethics committee [29 April 2022—N 0014840/22]. All included patients were positive for the SARS-CoV-2 RT-PCR test on nasopharyngeal and oropharyngeal swabs. The guidelines approved by the Italian Medicines Agency (AIFA) in December 2020 during the Delta wave and the updated version of October 2021 during the Omicron wave were followed for in-hospital treatment [8]. As to outcomes, we considered the need for admission to an intensive care unit (ICU) for at least one day and all-cause, 30-day mortality. 

Patients were grouped according to their vaccination status and the prevalent variant at the time of enrollment. Participants were classified as unvaccinated if they had never received a COVID-19 vaccine, while vaccinated patients were considered to be all those patients who had undergone at least one shot of the vaccine. These patients were further classified according to the number of doses received.

### Statistical Analysis

Data are presented as mean ± standard deviation for continuous variables and as percentages for categorical variables. For categorical variables, comparison between groups was performed using the chi-square test or the Fisher’s exact test where appropriate. In the case of continuous variables, the comparison between groups was performed using the Student *t*-test for independent samples. Instead, the comparison between multiple groups was performed with the ANOVA test. As a post hoc test, we applied Bonferroni’s multiple comparison test. We performed multiple logistic regression using vaccination status as the independent variable to estimate the impact on outcomes adjusting for age, gender, and any variable with a *p*-value less than 0.01 in the univariate analysis. Additional multiple logistic regression models were run using the virus variant as the independent variable, with adjustment for multiple confounders. Results of the multivariate analysis were reported using OR and 95% CI. Statistical analysis was performed using IBM SPSS version 20.0.

## 3. Results

The current analysis includes 821 patients (mean age 62 ± 18 years), of whom 59% were men. Overall, 545 of the patients (66%) tested positive during the Delta variant wave and 276 (34%) during the Omicron variant wave (Table 1). 

### 3.1. Patients during Delta and Omicron Variant Waves

The mean age of patients during the Omicron wave was significantly higher than that during the Delta wave (67 vs. 59 years, *p* < 0.0001), and patients presented with a greater number of comorbid conditions, mostly hypertension and chronic obstructive pulmonary disease (see Table 2). Tocilizumab, corticosteroids, and anticoagulants were used more frequently during the Delta wave than during the Omicron wave. At univariate analysis, no differences were found in 30-day mortality and the need for ICU admission between the two waves (Table 2). Figure 1 shows the differences in ICU admission and mortality rate between the two variants, compared to the overall numbers. 

### 3.2. Vaccination Status

As shown in Table 1, 358 patients (44%) were vaccinated, 7% with a single dose, 28% with two doses, and 9% of patients had also received a booster dose. The proportion of patients vaccinated was much higher during the Omicron variant wave than during the Delta variant wave (57% vs. 37%, respectively). In addition, during the Omicron variant wave, 27% of the patients had received a booster dose (vs. 1% among patients during the Delta variant wave). As illustrated in Figure 2, Pfizer-Biontech accounted for 40% of all vaccines, followed by Moderna (29%) and Astra-Zeneca (12%).

### 3.3. Vaccinated and Unvaccinated Patients

Table 1 illustrates that the mean age of vaccinated patients was significantly higher (68 vs. 57 years, *p* < 0.001). Accordingly, the proportion of patients with at least two comorbid conditions was twice as high among vaccinated ones (62 vs. 31%), and that was true for all conditions considered. At presentation, PaO_2_/FiO_2_ ratio in unvaccinated patients was significantly lower than among vaccinated ones. Tocilizumab use was three times higher among unvaccinated patients, while the use of monoclonal antibodies was three times higher among vaccinated patients.

### 3.4. Different Vaccination Regimens

Table 3 compares unvaccinated patients with those vaccinated with different doses (one, two, or three). Patients who had received just one dose appeared to be very similar to unvaccinated patients. Instead, patients who had received at least two doses were approximately 20 years older, and the rate of comorbidity was twice as high compared to unvaccinated patients or those who had received just one dose. That was true for all cardiovascular conditions, diabetes, chronic kidney disease, and Alzheimer’s disease.

Nearly one out of four unvaccinated patients, or those who had received just one dose, were treated with tocilizumab (24 and 22%, respectively), while its use was uncommon among patients vaccinated with more than one dose. Conversely, among the latter patients, there was a five-to-six-fold increase in the use of monoclonal antibodies compared to unvaccinated patients and those with just one dose of vaccine.

### 3.5. Clinical Severity at Emergency Room Presentation by Vaccination Status

At presentation, 586 patients (71%) had interstitial pneumonia, and 3% were diagnosed with pulmonary embolism. Of the 821 patients, 15% were directly discharged, and 65% were admitted to low-to-medium intensity medical wards.

Among unvaccinated patients, 14% were discharged home from the emergency department (ED), while 24% required at least one day in the ICU. As for patients with one vaccine dose, 27% did not require hospitalization, 61% were admitted to a medical ward, and 12% were transferred to the ICU. In the group of patients with two vaccine doses, 11% were discharged from ED, 72% required admission to a medical ward, and 17% were admitted to ICU. Finally, 20% of patients with three vaccine doses were discharged home from ED, 70% were admitted to a medical ward, and 10% required admission to ICU.

Pneumonia was diagnosed in 74% of unvaccinated patients, in 63%, 73%, and 57% of patients with one, two, or three vaccine doses, respectively. Pulmonary embolism was diagnosed in 4% of unvaccinated patients and only in 4 vaccinated patients (2 in single-dose patients and 2 in patients vaccinated twice).

### 3.6. Outcomes by Vaccination Status

Overall, 20% of the patients were admitted to ICU. A total of 113 (24%) unvaccinated patients required at least 1 day of ICU stay, compared to 12%, 17%, and 11% of patients vaccinated with one, two, or three doses, respectively (Figure 3).

Non-invasive ventilation was used in 8% of cases, high-flow-nasal-cannula (HFNC) in 13%, while 9% of the patients required either intubation or tracheostomy. The all-cause mortality rate at 30 days was 14% (Figure 4). A total of 58 (13%) unvaccinated patients, 2% of those vaccinated with a single dose, 17% of those with 2 doses, and 22% with a booster dose, died within 30 days of ED admission. Table 3 documents the occurrence of the outcomes among unvaccinated patients and those vaccinated with different numbers of doses. 

### 3.7. Impact of Vaccination on Outcomes

After adjusting for several confounding factors, unvaccinated patients had a higher probability of being admitted to the ICU than patients who received a vaccine (OR 2.0, 95% CI 1.3–3.1). Such protection was most evident among patients who had received more than one dose of vaccine [two doses OR 1.9, 95% CI 1.1–3.0; three doses OR 3.6, 95% CI 1.5–8.5] (see Table 4). The 30-day mortality rate was higher among unvaccinated people than those vaccinated twice (OR 1.9, 95% CI 1.5–2.9) or at least once (OR 1.7, 95% CI 1.3–2.7).

### 3.8. Impact of Variant on Outcomes

To evaluate rates of mortality and admission to the ICU during the two wave periods, we ran a multivariate logistic regression adjusted for age, gender, and additional confounders. As shown in Table 5, there was no difference in mortality at 30 days in the two waves of COVID-19 (OR 0.6, 95% CI 0.3–1.0). In contrast, the need for admission to the intensive care unit was significantly associated with the Delta variant (OR 1.9, 95% CI 1.2–3.1).

## 4. Discussion

In our study of COVID-19-positive patients who attended the Emergency Room during the dominant Delta or Omicron variant in Italy, we have documented that during the Omicron wave, patients were significantly older and had a greater number of medical comorbidities. Also, the vaccination rate was much higher than during the Delta variant wave. This is probably explained by higher vaccination adherence among older individuals. Data at that time would support that vaccines have a higher efficacy against the Delta variant [9].

Non-vaccinated patients had a lower mean age than those who had received at least one dose of vaccine [10]. On the one hand, younger people were not the initial target of vaccination campaigns [11], and when vaccines became accessible to everybody regardless of age, there was widespread hesitancy due to the report of possible serious adverse reactions, especially in younger individuals [12,13,14,15]. The age difference between vaccinated and unvaccinated patients explains why the prevalence of chronic comorbid conditions, as well as medication use, was higher among those who were vaccinated [16]. For instance, the use of angiotensin-converting enzyme inhibitors/angiotensin receptor blockers (ACEi/ARBs) was more common, paralleled by a higher prevalence of arterial hypertension and heart failure. Yet, at initial presentation, patients who were not vaccinated were in worse clinical condition and had more severe pulmonary involvement consistent with the available literature [9,17,18,19]. Vaccinated patients had a greater number of traditional risk factors for developing severe forms of COVID-19, and as such, they received antiviral therapies more frequently than those who were unvaccinated. 

Patients who had received two or three doses of vaccine were less likely to be admitted to ICU compared to unvaccinated patients. Patients who had been vaccinated once or twice also presented a lower 30-day mortality, while the same was not true for those who had received a booster dose. This can be explained by choice of outcome and a higher overall non-viral-related mortality among patients indicated to receive a booster dose. In our study, we could not evaluate the actual causes of death to judge whether COVID-19 played a causative role or was only a precipitating factor [17]. 

Several studies have documented the protective effect afforded by vaccination against serious forms of COVID-19. In a large epidemiological study conducted in England, both mRNA and adenoviral vector vaccines were associated with reduced mortality and admission to ICU among patients infected with Delta and Omicron virus variants. However, this study could not consider some possible confounding factors, such as the administration of antiviral therapies and monoclonal antibodies [20].

Another prospective study in Delta and Omicron virus variant infected patients has similarly shown that vaccines are extremely effective at preventing severe forms of COVID-19, especially in elderly patients with multiple comorbidities [21]. Interestingly, vaccines appeared to be less potent against the Omicron virus variant than they were against the Delta variant [22]. This finding is at odds with our results, but it is probably due to a different choice of outcomes. Indeed, the authors did not consider ICU admission or mortality as in our study, but rather the need for oxygen treatment with a fraction of inspired oxygen (FiO_2_) > of 28% and the use of any form of positive pressure support during ventilation. Therefore, the Omicron variant seems associated with milder disease when evaluated with different severity indexes [22].

Relative to the additional benefit of a booster dose, available data is consistent with the findings of our study: a population study conducted in Thailand documented a further protective effect of the booster dose during both Delta and Omicron variant waves. The extent of such protection was higher than that in our study, but this difference is probably explained by the fact that we included patients presenting to an emergency department and, thus, in more severe clinical conditions [23].

In a small subset of patients for whom information was available, influenza vaccination was associated with a reduced mortality rate, as had already been reported [24,25]. Influenza vaccination likely elicits antiviral protection not limited to its target [26,27], thus boosting immune protection even in elderly persons [28]. Also, it is noteworthy that individuals adhering to COVID-19 but also to influenza vaccination campaigns are more compliant. Likewise, they are more likely to refer to the hospital earlier, in less severe condition, than those who do not [29,30].

## 5. Limitations

The present study has some limitations. As for any observational study, selection bias is an inherent risk. Criteria for hospitalization, for instance, were not standardized and were based mainly on clinical judgment; thus, there may have been differences between the first and second waves. A similar problem may have presented for ICU admission. In addition, viral variants were not identified by molecular methods but adjudicated based on the variant prevalent in Italy during the period of interest (further information can be found at [31]). In this respect, we cannot exclude conclusively that some patients might have been wrongly classified. Also, it is worth noting that antigenic testing had a different sensitivity against the two variants, being specifically less sensitive to the Omicron variant [32]. While patients did not only refer to the hospital because of a positive antigen swab, it might have created a selection bias. Indeed, patients with mild symptoms might also have waited longer to refer to the hospital because of a negative antigen test.

We also lack information about the lag time between vaccination and COVID-19 positivity. Indeed, several studies seem to agree that vaccine efficacy declines with time, but we could not estimate the impact in our study. Yet, it is worth noting that persons who were infected with the Omicron variant were more likely to have had received their shot far before those who were infected with the Delta variant, increasing their likelihood of developing an infection [33]. However, prior data suggest that the waning of vaccine efficacy over time seems to mainly affect protection from infection rather than the severity of the disease [34]. Thus, it is likely that the lack of such information did not significantly impact the findings of our study.

Finally, it is important to remember that our paper concentrates on the Delta and Omicron variants; thus, it does not apply to all those countries experiencing other variants, such as XBB, a recombinant virus and progeny of BA.2.75 and BA.2.10.

## 6. Conclusions

In patients hospitalized for COVID-19, vaccination against SARS-CoV-2 appears to provide significant protection. The protective effect on ICU admission is seen only after two doses of the mRNA vaccine and is independent of the viral variant, while mortality appears to be reduced by one or two vaccinations but not by a booster dose. As information about vaccines and COVID-19 continues to evolve, further research is needed to confirm protection against other variants and to test new vaccines, different vaccination schedules, and regimens as well as routes.

## Figures and Tables

**Figure 1 vaccines-11-00373-f001:**
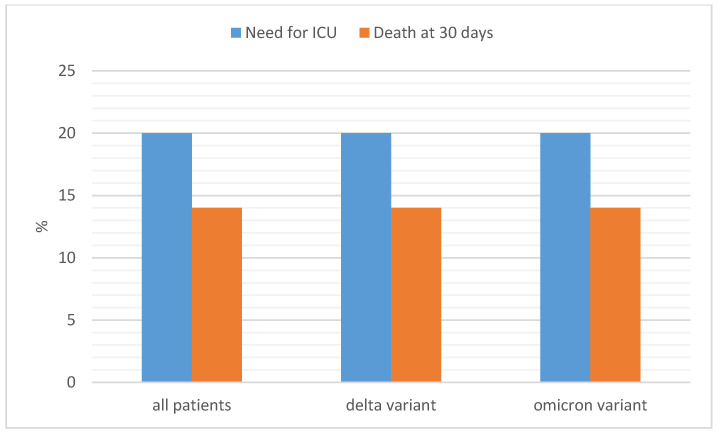
Need for intensive care unit (ICU) admission and all-cause 30-day mortality by virus variant.

**Figure 2 vaccines-11-00373-f002:**
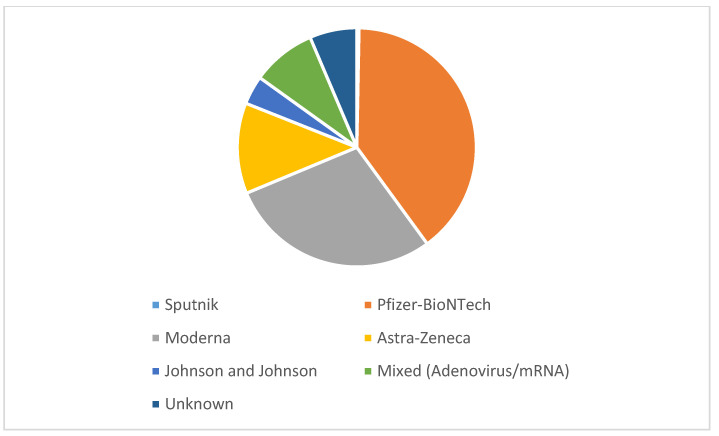
Types of COVID-19 vaccine received (relative percentages).

**Figure 3 vaccines-11-00373-f003:**
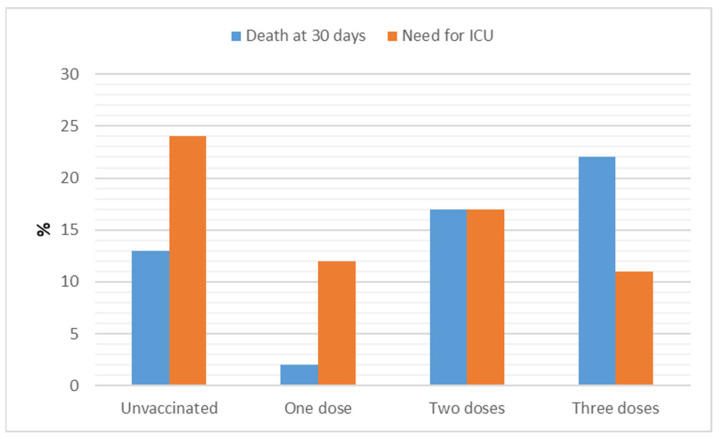
Need for intensive care unit (ICU) admission and all-cause 30-day mortality by doses of vaccine received. *p* < 0.005 among groups considering ICU admission; *p* < 0.01 among groups considering death at 30 days.

**Figure 4 vaccines-11-00373-f004:**
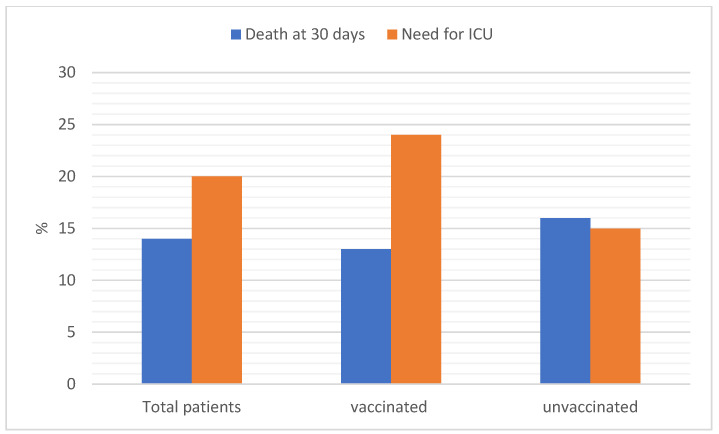
Need for intensive care unit (ICU) admission and all-cause 30-day mortality by vaccination status. *p* < 0.001 comparing ICU admission in vaccinated vs. unvaccinated patients.

**Table 1 vaccines-11-00373-t001:** COVID-19, patients’ demographic, comorbidities, laboratory, and outcomes data.

	All Patients (N = 821)	Unvaccinated (N = 463)	Vaccinated (N = 358)	*p*
**Demographic data**				
Age (years, mean value ± SD)	62 ± 18	57 ± 17	68 ± 18	**<0.0001**
Males N (%)	485 (59)	276 (59)	209 (58)	0.1270
BMI (mean value ± SD)	28 ± 6	28 ± 6	27 ± 6	0.5325
Delta variant N (%)	545 (66)	345 (75)	200 (56)	**<0.0001**
Omicron variant N (%)	276 (33)	118 (25)	158 (44)	**<0.0001**
**Comorbidities** N (%)				
Diabetes	116 (14)	54 (12)	62 (17)	**0.0211**
Hypertension	324 (40)	140 (30)	184 (51)	**<0.0001**
Coronary Heart Disease	76 (9)	19 (4)	57 (16)	**<0.0001**
Congestive Heart Failure	55 (7)	16 (3)	39 (11)	**<0.0001**
Cardiac Valve Disease	14 (2)	6 (1)	8 (2)	0.3029
Atrial Fibrillation	83 (10)	29 (6)	54 (15)	**<0.0001**
COPD	77 (9)	23 (5)	54 (15)	**<0.0001**
Active cancer	99 (12)	32 (7)	67 (19)	**<0.0001**
Other lung conditions *	56 (7)	36 (8)	20 (6)	0.2173
Chronic Kidney Disease	50 (6)	14 (3)	36 (10)	**<0.0001**
Parkinson disease	13 (2)	3 (1)	10 (3)	**0.0213**
Alzheimer’s disease	38 (5)	12 (3)	26 (7)	**0.0016**
Obesity	341 (41)	187 (41)	154 (43)	0.4486
Other Chronic Diseases **	41 (5)	17 (4)	24 (7)	**0.0479**
N. of comorbidities > 1	361(44)	144 (31)	217 (62)	**<0.0001**
**At-home treatment** N (%)				
Anticoagulants	112 (14)	54 (12)	58 (16)	0.0603
ACEi/ARB	184 (22)	81 (17)	103 (29)	**0.0001**
**Laboratory Values** **(Mean value ± SD)**				
BUN (mg/dL)	26 ± 24	24 ± 25	29 ± 24	**0.0042**
LDH (UI/L)	333 ± 299	365 ± 219	292 ±160	**<0.0001**
CRP (mg/L)	70 ± 70	66 ± 67	75 ± 75	0.0689
Procalcitonin (ng/mL)	3 ± 24	3 ± 31	2 ± 11	0.2739
Neutrophils (×10^9^/L)	6326 ± 4267	6110 ± 3610	6310 ± 3270	0.7472
Eosinophils (×10^7^/L)	72 ± 265	58 ± 199	95 ± 329	**0.0269**
Lymphocytes (×10^9^/L)	1222 ± 757	1157 ± 629	1305 ± 890	**0.0081**
D-dimer (ng/mL)	2330 ± 4995	2185 ± 4925	2524 ± 5088	0.3608
Fibrinogen (mg/dL)	493 ± 164	497 ± 159	488 ± 171	0.4349
PaO_2_/FiO_2_	302 ± 106	287 ± 102	320 ± 107	**<0.0001**
**In-hospital treatment** N (%)				
Anticoagulants	568 (69)	335 (72)	229 (64)	**0.0102**
Corticosteroids	568 (69)	346 (74)	222 (62)	**<0.0001**
Remdesivir	221 (27)	147 (32)	74 (21)	**0.0004**
Monoclonal Antibodies ***	70 (8)	20 (4)	50 (14)	**<0.0001**
Tocilizumab	134 (16)	106 (23)	28 (8)	**<0.0001**
Other anti-IL-6	41 (5)	23 (5)	18 (5)	0.9686

* Pulmonary fibrosis, asthma, obstructive sleep apnea. ** Inflammatory bowel diseases, autoimmune disorders, chronic neurological disorders. *** casirivimab and imdevimab. COPD: Chronic obstructive pulmonary disease. BUN: Blood Urea Nitrogen, LDH: lactic dehydrogenase, CRP: C-reactive protein. Significant *p* values are in bold.

**Table 2 vaccines-11-00373-t002:** COVID-19, patients’ demographic, comorbidities, laboratory, and outcomes data in groups with different predominant variants.

	All Patients (N = 821)	Delta Variant (N = 545)	Omicron Variant (N = 276)	*p*
**Demographic data**				
Age (years, mean value ± DS)	62 ± 18	59 ± 19	67 ± 17	**<0.0001**
Male sex N (%)	485 (59)	319 (59)	166 (60)	0.6571
BMI (mean value ± DS)	28 ± 6	28 ± 6	28 ± 7	0.9614
**Comorbidities** N (%)				
Diabetes	116 (14)	74 (14)	42 (15)	0.5241
Hypertension	324 (40)	184 (34)	140 (51)	**<0.0001**
Coronary Heart Disease	76 (9)	44 (8)	32 (12)	0.1001
Chronic Heart Failure	55 (7)	32 (6)	23 (8)	0.1826
Cardiac Valve Disease	14 (2)	9 (2)	5 (2)	0.8670
Atrial Fibrillation	83 (10)	59 (11)	24 (9)	0.3389
COPD	77 (9)	43 (8)	34 (12)	**0.0398**
Active cancer	99 (12)	57 (10)	42 (15)	**0.0479**
Other lung conditions *	56 (7)	39 (7)	17 (6)	0.5926
Chronic Kidney Disease	50 (6)	27 (5)	23 (8)	0.0558
Parkinson disease	13 (2)	8 (2)	5 (2)	0.7697
Alzheimer disease	38 (5)	21 (4)	17 (6)	0.1373
Obesity	341 (41)	240 (44)	101 (37)	**0.0409**
Other Chronic Diseases **	41 (5)	25 (5)	16 (6)	0.4521
N. of comorbidities > 1	361(44)	213 (39)	148 (54)	**<0.0001**
**At-home treatment** N (%)				
Anticoagulants	112 (14)	78 (14)	34 (12)	0.4319
ACEi/ARB	184 (22)	105 (19)	79 (29)	**0.0024**
**Vaccine status** N (%)				
No vaccine	463 (56)	345 (63)	118 (43)	**<0.0001**
One shot	56 (7)	46 (8)	10 (4)	**0.0011**
Two shots	226 (28)	153 (28)	73 (26)	0.6226
Three shots	76 (9)	1 (1)	75 (27)	**<0.0001**
At least 1 shot	358 (44)	200 (37)	158 (57)	**<0.0001**
**Laboratory Values** **(Mean value ± DS)**				
BUN (mg/dL)	26 ± 24	24 ± 22	31 ± 29	**0.0005**
LDH (UI/L)	333 ± 299	346 ± 193	308 ± 208	**0.0119**
CRP (mg/L)	70 ± 70	69 ± 68	72 ± 74	0.6142
Procalcitonin (ng/mL)	3 ± 24	3 ± 28	2 ± 13	0.6812
Eosinophils (×10^7^/L)	72 ± 265	81 ± 317	54 ± 91	0.0671
Lymphocytes (×10^9^/L)	1222 ± 757	1216 ± 658	1232 ± 922	0.8030
D-dimer (ng/mL)	2330 ± 4995	2046 ± 4803	2883 ± 5315	**0.0407**
Fibrinogen (mg/dL)	493 ± 164	521 ± 167	438 ± 140	**<0.0001**
PaO_2_/FiO_2_	302 ± 106	304 ± 100	298 ± 117	0.4739
**In-hospital treatment** N (%)				
Anticoagulants	568 (69)	465 (85)	215 (78)	**0.0077**
Corticosteroids	568 (69)	401 (74)	167 (61)	**<0.0001**
Remdesivir	221 (27)	156 (29)	65 (24)	0.1180
Monoclonal Antibodies ***	70 (8)	43 (8)	27 (10)	0.3590
Tocilizumab	134 (16)	130 (24)	4 (1)	**<0.0001**
Other anti-IL-6	41 (5)	16 (3)	25 (10)	**<0.0001**

* Pulmonary fibrosis, asthma, obstructive sleep apnea. ** Inflammatory bowel disease, autoimmune disorders, chronic neurological disorders. *** casirivimab and imdevimab. COPD: Chronic obstructive pulmonary disease. BUN: Blood Urea Nitrogen, LDH: lactic dehydrogenase, CRP: C-reactive protein. Significant *p* values are in bold.

**Table 3 vaccines-11-00373-t003:** COVID-19, patients’ demographic, comorbidities, laboratory, and outcomes data.

	**Unvaccinated** **(N = 463)**	**One Dose** **(N = 56)**	**Two Doses** **(N = 226)**	**Three Doses** **(N = 76)**	** *p* **
**Demographic data**					
Age (years, mean value ± SD)	57 ± 17	52 ± 18	71 ± 16	74 ± 15	**<0.0001**
Male sex N (%)	276 (59)	35 (63)	127 (56)	47 (62)	0.6931
BMI (mean value ± SD)	28 ± 6	28 ± 5	27 ± 5	28 ± 8	0.5694
Delta variant N (%)	345 (75)	46 (88)	153 (68)	1 (1)	**<0.0001**
Omicron variant N (%)	118 (25)	10 (18)	73 (32)	75 (99)	**<0.0001**
**Comorbidities** N (%)					
Diabetes	54 (12)	4 (7)	45 (20)	13 (17)	**0.0081**
Hypertension	140 (30)	16 (28)	125 (55)	43 (57)	**<0.0001**
Coronary Heart Disease	19 (4)	4 (7)	39 (27)	14 (18)	**<0.0001**
Chronic Heart Failure	16 (3)	0 (0)	30 (13)	9 (12)	**<0.0001**
Cardiac Valve Disease	6 (1)	0 (0)	6 (3)	2 (3)	0.6317
Atrial Fibrillation	29 (6)	4 (7)	39 (17)	111 (14)	**<0.0001**
COPD	23 (5)	3 (5)	39 (17)	12 (16)	**<0.0001**
Active cancer	32 (7)	4 (7)	49 (22)	14 (18)	**<0.0001**
Other lung conditions *	36 (8)	3 (5)	14 (6)	3 (4)	0.0586
Chronic Kidney Disease	14 (3)	1 (2)	23 (10)	12 (16)	**<0.0001**
Parkinson disease	3 (1)	1 (2)	6 (3)	3 (4)	0.0674
Alzheimer disease	12 (3)	4 (7)	11 (5)	11 (14)	**<0.0001**
Obesity	187 (41)	27 (45)	98 (43)	29 (38)	0.3854
Other Chronic Disease **	17 (4)	4 (7)	15 (7)	5 (7)	0.8533
N. of comorbidities > 1	144 (31)	17 (30)	147 (65)	53 (70)	**<0.0001**
**At-home treatment** N (%)					
Anticoagulants	54 (12)	3 (5)	44 (19)	11 (14)	**0.0080**
ACEi/ARB	81 (17)	13 (23)	65 (29)	25 (33)	**0.0021**
**Laboratory Values** **(Mean value ± Sd)**					
BUN (mg/dL)	24 ± 25	20 ± 12	29 ± 25	36 ± 24	**<0.0001**
LDH (UI/L)	365 ± 219	327 ± 164	298 ± 169	269 ± 122	**<0.0001**
CRP (mg/L)	66 ± 67	77 ± 86	80 ± 74	64 ± 69	0.1428
Procalcitonin (ng/mL)	3 ± 31	1 ± 2	1 ± 3	5 ± 22	0.4610
Eosinophils (×10^7^/L)	58 ± 199	181 ± 729	109 ± 186	73 ± 126	**0.0091**
Lymphocytes (×10^9^/L)	1157 ± 629	1463 ± 867	1290 ± 861	1315 ± 985	**0.0174**
D-dimer (ng/mL)	2185 ± 4925	2076 ± 5212	2746 ± 5449	2137 ± 3140	0.6138
Fibrinogen (mg/dL)	497 ± 159	513 ± 212	505 ± 168	426 ± 133	**0.0033**
PaO_2_/FiO_2_	287 ± 102	330 ± 88	314 ± 110	329 ± 113	**<0.0001**
**In-hospital treatment** N (%)					
Anticoagulants	390 (84)	40 (70)	190 (84)	60 (79)	0.1732
Corticosteroids	346 (74)	35 (61)	144 (64)	43 (57)	**0.0009**
Remdesivir	147 (32)	13 (23)	47 (21)	14 (18)	**0.0047**
Monoclonal Antibodies ***	20 (4)	1 (2)	35 (15)	14 (18)	**<0.0001**
Tocilizumab	106 (23)	12 (21)	15 (7)	1 (1)	**<0.0001**
Other anti-IL-6	23 (5)	5 (9)	10 (4)	3 (4)	0.5954

* Pulmonary fibrosis, asthma, obstructive sleep apnea. ** Inflammatory bowel disease, autoimmune disorders, chronic neurological disorders. *** casirivimab and imdevimab. COPD: Chronic obstructive pulmonary disease. BUN: Blood Urea Nitrogen, LDH: lactic dehydrogenase, CRP: C-reactive protein. Significant *p* values are in bold.

**Table 4 vaccines-11-00373-t004:** Multiple logistic regression analysis to evaluate differences in outcomes (need for ICU and death at 30 days) on the bases of vaccine status (no vaccination, one dose, two doses, three doses, at least one dose) after correction for age, sex, comorbidities, use of anticoagulants, and treatment. (Significant *p* values are in bold).

	Unvaccinated	One Dose	*p*	*Odds Ratio*	*95%CI*
Need for ICU admission	113/463 (24)	7/56 (12)	0.153	1.9	0.8–4.4
All-cause mortality at 30 days	58/463 (13)	1/56 (1)	**0.048**	3.9	1.8–19.0
	**Unvaccinated**	**two doses**			
Need for ICU admission	113/463 (24)	39/226 (17)	**0.012**	1.9	1.1–3.0
All-cause mortality at 30 days	58/463 (13)	38/226 (17)	**0.049**	1.9	1.5–2.9
	**Unvaccinated**	**three doses**			
Need for ICU admission	113/463 (24)	8/76 (11)	**0.004**	3.5	1.5–8.5
All-cause mortality at 30 days	58/463 (13)	17/76 (22)	0.489	1.3	0.6–2.8
	**Unvaccinated**	**At least one dose**			
Need for ICU admission	113/463 (24)	54/358 (15)	**0.002**	2.0	1.3–3.1
All-cause mortality at 30 days	58/463 (13)	56/358 (16)	**0.047**	1.7	1.3–2.7

**Table 5 vaccines-11-00373-t005:** Multiple logistic regression analysis to evaluate differences in outcomes (need for ICU and death at 30 days) based on different variants (Delta or Omicron) after correction for age, sex, number of comorbidities, main chronic diseases, vaccine status, and treatment. (Significant *p* values are in bold).

	Delta	Omicron	*p*	*Odds Ratio*	*95%IC*
Need for ICU admission	111	56	**0.007**	1.9	1.2–3.1
All-cause mortality at 30 days	74	40	0.064	0.6	0.3–1.0

## Data Availability

Data are available from the corresponding author upon reasonable request.

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
