# Peer review of "Effect of COVID-19 Vaccination on the In-Hospital Prognosis of Patients Admitted during Delta and Omicron Waves in Italy"

_vaccines, 2023, doi:10.3390/vaccines11020373_

Round 1
Reviewer 1 Report
There are some missing information and gaps in the present report.
Specific comments:
1. As per the instructions to authors, the abstract should be a total of about 200 words maximum. The abstract should be a single paragraph and should follow the style of structured abstracts but without the headings.
2. Please standardize the stylization of COVID-19 as 'COVID-19' instead of having 'COVID19', Covid-19' or 'covid'.
3. In the introduction section, it is also relevant to mention that a recent social media analysis found that a topic of public interest was the public concerns’ regarding the new Delta and Omicron variants and the potential lack of efficacy of the ancestral COVID-19 vaccines against these new variants (citation: pubmed.ncbi.nlm.nih.gov/36146535).
4. Please change "Materials and Methods" to "Methods".
5. Please change "pneumonia on X-ray" to "pneumonia on plain chest radiograph".
6. What counts as "vaccinated" versus "unvaccinated"? Some countries now consider primary immunisation for COVID-19 to require 3 doses of mRNA vaccine and many individuals have received their booster shots. This distinction is important.
7. Moving from Delta to Omicron, the criteria of patients requiring hospitalization may have changed as well as more countries now manage COVID-19 patients in the community. This may have affected the results and your interpretations.
9. "... admitted to the Emergency Medicine" - do you mean admitted to the hospital or does this also include short overnight stays in the treatment units?
10. Please change "would testify that vaccines" to "would support that vaccines".
11. Moving from Delta to Omicron, lateral flow tests have become the main tool used for detecting COVID-19, with PCR tests mainly limited to hospital settings, and there are studies suggesting a drop in antigen test sensitivities with the Omicron variant (citation: pubmed.ncbi.nlm.nih.gov/36431077). This is relevant to mention as these testing delays may also affect the timing when people refer themselves to the hospitals.
12. "... adjudicated based on the variant prevalent in Italy during the period of interest" - are there published studies or surveillance reports from the official health ministry to support these assumptions? At least some citations are necessary.
13. "We lack information about the lag time between vaccination and Covid-19 positivity" - the more significant issue here may be that for patients who were infected by the Omicron variant, they received their vaccinations quite some time ago and their antibody protection may have waned significantly, affecting their illness vulnerability (citation: pubmed.ncbi.nlm.nih.gov/35248996).
14. Information presented in this report may also be less pertinent now that many countries are way beyond BA.1 now. It was BA.2 and then BA.4/5 and now many countries are being hit by XBB (which is a recombinant virus and progeny of BA.2.75 and BA.2.10).
15. Please change "Delta and Omicron viral variant" to "Delta and Omicron variants".
Author Response
There are some missing information and gaps in the present report.
Thank you for your comments, we have addressed them below.
Specific comments:
- As per the instructions to authors, the abstract should be a total of about 200 words maximum. The abstract should be a single paragraph and should follow the style of structured abstracts but without the headings.
Thank you for your observation. We have shortened the abstract and eliminated the headings.
- Please standardize the stylization of COVID-19 as 'COVID-19' instead of having 'COVID19', Covid-19' or 'covid'.
We have done as you suggested.
- In the introduction section, it is also relevant to mention that a recent social media analysis found that a topic of public interest was the public concerns’ regarding the new Delta and Omicron variants and the potential lack of efficacy of the ancestral COVID-19 vaccines against these new variants (citation: pubmed.ncbi.nlm.nih.gov/36146535).
Thank you for your suggestion, we have added it to the text.
- 4. Please change "Materials and Methods" to "Methods"
We have done as you suggested.
- Please change "pneumonia on X-ray" to "pneumonia on plain chest radiograph".
We have done as you suggested
- What counts as "vaccinated" versus "unvaccinated"? Some countries now consider primary immunisation for COVID-19 to require 3 doses of mRNA vaccine and many individuals have received their booster shots. This distinction is important.
We use the term vaccinated to include all those people who received at least one dose of the vaccine. We have added explicated it in the text too (“…vaccinated patients were considered to be all those patients who had undergone at least one shot of the vaccine.)
- Moving from Delta to Omicron, the criteria of patients requiring hospitalization may have changed as well as more countries now manage COVID-19 patients in the community. This may have affected the results and your interpretations.
Thank you for your observation. We have added it to the limitations section.
- "... admitted to the Emergency Medicine" - do you mean admitted to the hospital or does this also include short overnight stays in the treatment units?
With this statement we intend all patients who entered the Emergency Room, without taking into consideration whether or not they were hospitalized or how long their stay lasted. We have changed the sentence to “… admitted to the Emergency Room” to clarify it.
- Please change "would testify that vaccines" to "would support that vaccines".
We have done as you suggested
- Moving from Delta to Omicron, lateral flow tests have become the main tool used for detecting COVID-19, with PCR tests mainly limited to hospital settings, and there are studies suggesting a drop in antigen test sensitivities with the Omicron variant (citation: pubmed.ncbi.nlm.nih.gov/36431077). This is relevant to mention as these testing delays may also affect the timing when people refer themselves to the hospitals.
Thank you for your comment, we have discussed this in our limitations section
- "... adjudicated based on the variant prevalent in Italy during the period of interest" - are there published studies or surveillance reports from the official health ministry to support these assumptions? At least some citations are necessary.
We have added a citation.
- "We lack information about the lag time between vaccination and Covid-19 positivity" - the more significant issue here may be that for patients who were infected by the Omicron variant, they received their vaccinations quite some time ago and their antibody protection may have waned significantly, affecting their illness vulnerability (citation: pubmed.ncbi.nlm.nih.gov/35248996).
Thank you for your suggestion, we have further discussed the issue
- Information presented in this report may also be less pertinent now that many countries are way beyond BA.1 now. It was BA.2 and then BA.4/5 and now many countries are being hit by XBB (which is a recombinant virus and progeny of BA.2.75 and BA.2.10).
We have added your observation to the limitations section
- Please change "Delta and Omicron viral variant" to "Delta and Omicron variants".
We have done as you suggested
Reviewer 2 Report
This is an observational study of outcomes of hospital-admitted patients during the Delta and Omicron SARS-CoV-2 waves in Italy, observing primarily the outcomes of death and ICU admission on the account of vaccination status, comorbidities, etc. The study may be of interest to some of the readership of the journal Vaccines, but some improvements are in order first, ranging from minor to more serious. In general, tables were more carefully prepared that figures, the latter being rather superficially done in several instances.
Line 17: Materials ("s" missing)
Line 27: "at least" (character missing)
Line 75: cross out the comma after the word "and"
Figure 1 is not clear for readers because it has no y-axis with proper labeling. What do the values written directly on top of each bar mean? Number of patients or percentage? I would guess percentage given that the "All patients" category does not sum up the other two categories (Delta and Omicron), but this should be made clear. There is no discussion of Figure 1 in the main text either that would have brought more clarity about this.
Line 177: clarity needed! is it 53 or 70% ?
Figure 3: again, the same comment as for Figure 1, there is no y-axis labeling; it is not clear what readers are looking at.
Figure 4: what do those asterisks mark on this figure? Comparison with p<0.001 performed between what and what? This should be clarified in the figure caption rather than just randomly writing a "*P<0.001" on the field of the figure itself.
Line 257: cite the other study that you mention here
Line 258: "has similarly shown" (not "shows")
Line 273: as it had already been reported ("it had" is missing)
Line 286: I believe this is a limitation that you can easily fix. The vaccination date appears on the patient's EU vaccination certificate and the time of Covid-19 positivity is the date of the PCR or rapid test that was positive and thus confirmed Covid19 infection. At least at the time of hospital admission, such a test must have been done, if not even sooner. Patients who were vaccinated must have presented their vaccination history as part of their medical history record even if the country does not have a centralized list of individual vaccinations performed throughout the pandemic.
Lines 288-290: this claim must be supported by citations of already published studies on this matter
Throughout the manuscript, there are also quite a few acronyms that were not defined at the time of their first occurrence (e.g. ACEi/ARB)
Author Response
This is an observational study of outcomes of hospital-admitted patients during the Delta and Omicron SARS-CoV-2 waves in Italy, observing primarily the outcomes of death and ICU admission on the account of vaccination status, comorbidities, etc. The study may be of interest to some of the readership of the journal Vaccines, but some improvements are in order first, ranging from minor to more serious. In general, tables were more carefully prepared that figures, the latter being rather superficially done in several instances.
Thank you for your comments. We have addressed the single issues below.
Line 17: Materials ("s" missing)
We have eliminated the sentence
Line 27: "at least" (character missing)
We have corrected the typo
Line 75: cross out the comma after the word "and"
We have done as you asked
Figure 1 is not clear for readers because it has no y-axis with proper labeling. What do the values written directly on top of each bar mean? Number of patients or percentage? I would guess percentage given that the "All patients" category does not sum up the other two categories (Delta and Omicron), but this should be made clear. There is no discussion of Figure 1 in the main text either that would have brought more clarity about this.
We have modified the figure and we have added a short explanation in the text.
Line 177: clarity needed! is it 53 or 70% ?
It is 70%, we have corrected the text
Figure 3: again, the same comment as for Figure 1, there is no y-axis labeling; it is not clear what readers are looking at.
We have modified the figure following your suggestion
Figure 4: what do those asterisks mark on this figure? Comparison with p<0.001 performed between what and what? This should be clarified in the figure caption rather than just randomly writing a "*P<0.001" on the field of the figure itself.
We have modified the figure to make it clearer.
Line 257: cite the other study that you mention here
We have added the reference
Line 258: "has similarly shown" (not "shows")
We have done as you suggested
Line 273: as it had already been reported ("it had" is missing)
We have done as you suggested
Line 286: I believe this is a limitation that you can easily fix. The vaccination date appears on the patient's EU vaccination certificate and the time of Covid-19 positivity is the date of the PCR or rapid test that was positive and thus confirmed Covid19 infection. At least at the time of hospital admission, such a test must have been done, if not even sooner. Patients who were vaccinated must have presented their vaccination history as part of their medical history record even if the country does not have a centralized list of individual vaccinations performed throughout the pandemic.
Unfortunately, it is not possible for us to access the data on time of vaccination. Patients were not required to show their EU vaccination certificate when entering the ER and, even when they did, physicians often did not report the date of vaccination, as it did not change testing protocols.
Lines 288-290: this claim must be supported by citations of already published studies on this matter
We have added a citation
Throughout the manuscript, there are also quite a few acronyms that were not defined at the time of their first occurrence (e.g. ACEi/ARB)
We have corrected this issue
Reviewer 3 Report
Thank you for sharing the article on effects of COVID-19 vaccination on prognosis of hospitalisation admission during recent Delta and Omicron epidemics. The article is very well written. Here some minor comments that could help to improve the article:
L34: Consider changing the 95%CI 1.5-8.50 to 95%CI 1.5-8.5 for consistency.
L80-82: Consider adding the weblinks stated here to the list of references for consistency and better readability.
Table 1, Table 2, Table 3: Consider using the same number of decimal places for p-values throughout.
Figure 1, Figure 3, Figure 4: Consider adding axis titles for better readability.
Figure 2: Are relative frequencies/percentages presented here?
General question: Was it necessary to obtain written/verbal consent from participants. Please clarify in the manuscript despite you stated "not applicable" in L309.
Author Response
Thank you for sharing the article on effects of COVID-19 vaccination on prognosis of hospitalisation admission during recent Delta and Omicron epidemics. The article is very well written. Here some minor comments that could help to improve the article:
Thank you for your comments, we have addressed the single issues below
L34: Consider changing the 95%CI 1.5-8.50 to 95%CI 1.5-8.5 for consistency.
We have eliminated this sentence
L80-82: Consider adding the weblinks stated here to the list of references for consistency and better readability.
We have done as suggested
Table 1, Table 2, Table 3: Consider using the same number of decimal places for p-values throughout.
We have followed your suggestion
Figure 1, Figure 3, Figure 4: Consider adding axis titles for better readability.
We have followed your suggestion
Figure 2: Are relative frequencies/percentages presented here?
We have made the figure clearer in what it is presenting
General question: Was it necessary to obtain written/verbal consent from participants. Please clarify in the manuscript despite you stated "not applicable" in L309.
We have added that verbal consent was obtained when patients entered the ER
Round 2
Reviewer 1 Report
More accurately, you meant to say "In our study of COVID-19 positive patients who attended the Emergency Room".
Author Response
Thank you for your comment, we have addressed the issue